

**Nitrogen and Warming Control the Vegetation in Inner Mongolia Tourist Area**
Qiong Sun[1], Xiaobing Hu[2], Chi Zhang[1,3]
(1) Tourism Institute of Beijing Union University, Beijing, 100101, China
(2) School of Engineering, University of Warwick, Coventry, CV4 7AL, United Kingdom.
(3) Centre for Creative Computing, Bath Spa University, Corsham SN13 0BZ, UK
**ABSTRACT**
The global warming and atmospheric nitrogen deposition problem has become more and more serious
under the influence of human activities, and it has become one of the hot issues in this field, which will
have far-reaching impact on all kinds of vegetation, thus the functioning of the ecosystem will be
changed, which will be reflected in climate warming process. Inner Mongolia Autonomous Region is
mainly composed of desert grasslands, so the development and protection of vegetation has
considerable significance on the region. However, in the current environment of global warming, few
studies have been carried out on desert grassland plants. In this paper, an in-depth study was carried
out on the impact of warming and nitrogen addition on soil temperature, vegetation reproductive
phenology and vegetation community seed rain under natural conditions during five-year period from
2011 to 2015. During the experimental period, we found that soil temperature and soil moisture
decreased with the increase of soil layer, and warming obviously shortened the time of budding,
flowering and fruiting of vegetation. However, no significant effect was found on nitrogen addition.
Meanwhile, the impact of interaction effect of warming and nitrogen addition on seed rain was not
obvious, but the year difference of all relevant indicators was significant.

**Key words:** Inner Mongolia; desert grassland; warming; nitrogen addition; vegetation
**INTRODUCTION**
Since 2000, with the increase in population, rapid economic development and increasing human
activities, there appear global environmental changes, including global warming, carbon dioxide
concentration increase, global nitrogen deposition (Zaldívar & Sanz, 2014; Oliva et al., 2014), which to
some extent, affected the original ecosystem of various species (Buendia et al., 2015; Liu et al., 2014).
The United Nations Intergovernmental Panel on Climate Change summarized on its fourth work report in
2007 that in the past few decades, earth's surface temperature increased by about 0.6 degrees Celsius.
Meanwhile, it was estimated that average global temperature would increase by a minimum of 1.8
degrees Celsius in the next 2100 years. With the harsh temperature changes, nitrogen settlement of the
Earth's surface is also facing a grim situation, which increased by at least 1 time. Moreover, estimated
based on the model of Elisabeth, in the next few decades, with the increase of various nitrogen
complexes with different activities, nitrogen settlement amount will continue to increase (Lin et al., 2011).
The rising temperature and nitrogen settlement has great impact on the growth of plants. It will change
the relationship between the various species, and thus affect the vegetation. Apart from the impact on



community, reproductive phenology and seeds of vegetation, it will also cause indirect effects on plants
through changing soil temperature, soil moisture and spring snowmelt time. There are many scholars
who had studied in these areas, and these to some extent appeal to human to get the warning that we
should protect the environment. For example, Xiaocheng Wen and Guangxin Lu put forward that
warming and nitrogen addition and interaction effect of the two changed the height of alpine grassland
plant communities and the fresh weight of aboveground biomass (Wen & Lu, 2015). Hongyu Guo, Wei
Wang and Guangxin Lu (2015) studied on how plant growth was influenced by warming and nitrogen
precipitation increase. While Jianan Cui, Yuexi Chen and Hui Sun (2015) studied on how warming and
nitrogen addition affected the mineral elements release of sequoia fresh litter. As one of the necessary
nutrients for the growth of plants and a kind of important fertilizer in the earth, nitrogen has drawn
extensive attention from experts. Abbasi et al. performed a study to determine the nitrogen-release
potential of residues added to soil and indicated that nitrogen concentration was an indicator for the
mineralization of organic residues (Abbasi et al., 2015). Yu and Jia observed the increase of soil organic
carbon (SOC) and total nitrogen (TN) in the soil and revealed that root biomass could increase SOC and
TN (Yu & Jia, 2014). Scharenbroch et al. proved that nitrogen availability of soil could promote the
growth of trees (Scharenbroch et al., 2013).
This paper carries out an in-depth analysis on three aspects of soil temperature, vegetation reproductive
phenology and vegetation seed rain based on the results obtained from the experiments during 2011 -
2015. It analyzes what impact will be brought by man-made warming and nitrogen addition to plants
under natural conditions, so as to estimate the crisis which Inner Mongolia grassland ecosystem may
face when there is natural environmental warming and increase of amount of nitrogen precipitation. Also,
it can be taken as a warning to call upon humans that we should protect the environment.
**MATERIALS AND METHODS**
In this experiment, we selected a base located in south central part of Inner Mongolia Ordos Grassland,
with a total area of 33,000 acres and a large part of the base are natural grasslands (Johansson et al.,
2008). Then we chose an area of desert grasslands and farming-grazing zone as the base of this
experiment since this region is relatively local with regional characteristics. The desert grasslands in that
region belong to the grassland vegetation subtype with strongest xerophytism among temperate
grassland vegetation (Urano et al., 2000), so it is a very suitable base for this experiment.
We carried out a five-year experiment on Inner Mongolia Ordos Grassland from March 2011 to March
2015. Firstly, a warming control device was set up for the quadrat. In the five consecutive years, we
conducted a soil warming and nitrogen addition process. Take the soil with 10 cm, 20 cm and 30 cm
surface layer as samples and make comparison. Meanwhile, record the relevant data on vegetation
reproductive phenology changes during 2011- 2013. Also, analyze the seeds obtained by the 64 seed
rain collectors in the five years, and the following can be concluded based on the analysis on its
temperature and humidity.
According to the test project, a control test on simulated warming and nitrogen addition under wild
natural conditions was carried out. In this test, warming was taken as a simulated global climate change
factor and nitrogen addition was taken as a man-made interference factor of global nitrogen settlement.
The design was divided into four kinds of treatment experiments which were warming treatment (W),
nitrogen fertilizer treatment (F), warming and nitrogen fertilizer treatment (WF) and control treatment (





without warming or nitrogen fertilizer treatment, C) respectively. Each treatment was repeated six times
and 24 treatment plots were included. The designed area of each plot was 8 m$^2$ (4m×2 m) and the total
area of the test was 192 m$^2$. All treatment groups in the plot were equally distributed in order to avoid the
influence of direction and orientation on test results. Infrared heating method was adopted for simulated
warming in the test. The infrared radiator, with a length 1.5 m and a width of 18 cm, was hung 2.75 m
high above the ground through a steel pipe with a diameter of 4.6 cm. The maximum power of the
infrared radiator was 2250 watt and the gear was adjusted to 8-speed when used. In order to avoid test
errors caused by shade effect of infrared radiator as well as block effect of infrared ray on rainfall, false
radiators of the same shape and size were installed at the same position of the same height as the real
ones for treatment groups which didn't require warming. Ammonium nitrate, whose chemical name is
NH$_4$NO$_3$, was selected as the nitrogen fertilizer, with a nitrogen content of 33.7%. The application time of
nitrogen fertilizer was during the end of June and beginning of July when the rainy season approaches
annually.
The test project began from March 2011 and the infrared heating test never stopped or was interrupted
for even a single day all year round from the beginning to the duration of this test.
**RESULTS**
When only soil warming was conducted, temperature increased respectively by 1.29 °C, 0.83 °C, 0.80
°C and 0.71 °C; when only nitrogen addition was conducted, the temperature of the surface layer
decreased by 0.18 °C, soil temperature decreased by 0.05 °C, 0.03 °C and 0.01 °C respectively at 10
cm, 20 cm and 30 cm layer. However, when both warming and nitrogen addition were conducted, soil
temperature of each layer increased respectively by 1.11 °C, 0.78 °C, 0.77 °C and 0.70 °C.
Through repeated variance measurement, it is found that all of the four dominant species were affected
by warming and nitrogen addition in the three growing seasons during 2011-2013 in the experiment. And
for the time of duration of budding, flowering, fructification and reproductive growth of vegetation, the
shortening of average time is more obvious under the warming situation (figure 1); while the shortening
of average time under nitrogen addition situation is not obvious (figure 1). When under both warming and
nitrogen addition situation, the shortening of average time is less obvious than that under mere warming
situation (figure 1).
Through the observation on data results of the impact of warming and nitrogen addition and the
interaction of the two on budding time of various species during the three growing seasons from 2011 to
2013, it is found that impact of warming varies on different vegetation species, i.e. there exists the
specificity; while for the same vegetation, there presents the difference on impact of warming in different
years. Seen from table 1, the budding time of breviflora griseb, convolvulus ammannii and bassia
prostrate was shortened significantly due to warming process in 2011. While in 2012, the budding time of
breviflora griseb and allium tenuissimum was shortened significantly under warming treatment condition.
And in 2013, the budding time of allium tenuissimum and bassia prostrate was shortened significantly
due to warming treatment.
At the same time, we also analyzed the experimental data of flowering time of each species in each
growing season under both warming and nitrogen addition treatment during 2011-2013. The results
show that the response of flowering time of each species to warming is similar with that of budding time;



there is also specificity in response of each species to warming. For example, in 2011, the flowering time
of stipa breviflora and bassia prostrate was significantly reduced under the influence of warming, but no
obvious effect was found on the flowering time of convolvulus ammannii and allium tenuissimum. In
2012, only the flowering time of allium tenuissimum was shortened obviously under the influence of
warming. While according to the data of 2013, warming has no significant effect on the flowering time of
all the species (table 2). Therefore, interaction effect of nitrogen addition and warming has little impact
on the flowering time of various species.
From the analysis of data results of fructification time of various species under both warming and
nitrogen addition treatment in the three growing seasons during 2011-2013, each species' response to
warming has the specificity, that is to say, in the same year, the results affected by warming have
different performance in different species. Similarly, for the vegetation itself, its response to warming
varies from year to year. In 2011, under the man-made warming treatment, the fructification time of stipa
breviflora, convolvulus ammannii and bassia prostrate was significantly shortened. In 2012, there was
significant shortening in fructification time of allium tenuissimum due to warming treatment; while seen
from table 3, fructification time of stipa breviflora was significantly shortened only by warming in 2013.
The response of each species to nitrogen addition conforms to the same law. There is no significant
effect on each species in each year.
From the analysis of data results of duration of reproductive growth of various species under both
man-made war ming and nitrogen addition treatment in the three growing seasons during 2011-2013, we
find that no matter what year it is, there is no significant effect of warming and nitrogen addition on
reproductive growth duration of convolvulus ammannii and allium tenuissimum. However, there is a year
difference on the effect of warming on the reproductive growth duration of stipa breviflora and bassia
prostrate. As can be seen from the table, in 2011 and 2012, the reproductive growth duration of stipa
breviflora was significantly shortened under the influence of warming; while in 2012, no significant effect
was found on the reproductive growth duration of stipa breviflora and bassia prostrate under the
influence of warming; in 2013, a significant shortening of the reproductive growth duration of bassia
prostrate was found, as seen in table 4.
During the five year period of growing seasons from 2011 to 2015, the 72 seed rain collectors on the
base collected altogether 147,757.19 seeds. The average density of the seed rain is 1229.6 ± 145.8
grains (Mean±SE), and all the seeds belong to 6 families, 8 genera and 8 species. In table 5, there is
detailed record of the seed rain information, including stipa breviflora, cleistogenes songorica and
wheatgrass, which are perennial bunch grass belonging to grass family. And stipa breviflora and
cleistogenes songorica are the vegetation which contribute most to seed rain with rate of contribution of
22.37 ± 2% and 13.11 ± 1% respectively. There are 2 families, 2 genuses and 2 species of perennial
herbs which are called allium tenuissimum and convolvulus ammannii. Among all the observed species,
allium tenuissimum (its contribution rate is 37.92 ± 2%) makes the greatest contribution to the quantity of
seeds rain, and the contribution rate of subshrub bassia prostrate is 14.86 ± 3%. As for annual herbs and
biennial herbs, their contribution rates are lower compared with those of the former ones and appear
randomly in different years.
Figure 2~6 show the impact of warming and nitrogen addition on seed rain density of cleistogenes
songorica, allium tenuissimum, bassia prostrate, stipa breviflora and plant community .




### DISCUSSION

Under the circumstances of global warming, in order to study the functional mechanism between
ecosystem and climate warming, various climate warming simulation tests were adopted, including
resistance heating, far-infrared irradiation, top-open and top-close field greenhouse (Choi et al., 2012),
intercross transplantation etc. In this study, warming through infrared radiation was carried out under wild
natural conditions. Through the test, we found that soil temperature increased while soil humidity
decreased after warming, suggesting that the warming test had a warming effect. However, the warming
effect and amplitude of this study is lower than that of the top-open warming method, which is mainly due
to the difference of warming methods. Far infrared warming method can keep the climatic factors of
sample plot such as natural air, wind speed, rainfall, temperature and humidity basically the same as the
natural climate conditions of the desert steppe, which minimizes the interference of human factors
caused by the warming method based on short-term control of air flow (Murakami et al., 2000), thus can
truly and objectively reflect and simulate the influence of climate warming on desert steppe.
Soil respiration (Nordgren et al., 2001) is a very complex soil ecological process. To some extent,
temperature is considered as the most important factor which influences total soil respiration rate.
However, in this study, no significant difference was found on soil respiration rate between warming
group and control group, no matter on seasonal change or on daily variation. For small area of sample
plot, the response of soil respiration to soil temperature changes showed great uncertainty, which may
be related to the spatial difference of water conditions, soil nutrient (Drechsel et al., 2001), biomass of
plants and microorganisms, etc, in addition to the short experimental time and low increasing extent of
temperature. Soil respiration is a very complicated biogeochemical process and not only the above
factors can affect directly or indirectly the generation and emissions of carbon dioxide in soil but also the
status and role of these factors will have corresponding changes with the change of temperature. Under
certain conditions, they may have modification, correction, and even cover-up effect on the effect of
temperature.
The far infrared warming test caused changes in dominant species and composition of the community
instead of changes of plant community. This may be because that far infrared warming effect, to some
extent, meets the demand for heat of the plant and changes the microclimate environment of plant
community and therefore influences the growth and development of plant to a certain degree (Hu et al.,
2015). Besides, warming has also changed the content of moisture in soil and influenced moisture
absorbing of plant. The growth and biomass of the plant are also affected to varying degrees. The results
showed that, for any kind of plant community under the environment of global warming, there are always
some species whose response to temperature rises are more sensitive, so as to destroy the interspecific
competition and cause changes in community dominant species and composition.
According to some experts, soil moisture decrease caused by warming can prevent the growth of
superficial roots, increase root mortality and enhance plant respiration, thus lower the net primary
productivity of plants. Warming changed the distribution of the underground biomass (Singh et al., 2014).
Compared with the control sample plot, underground biomass decreased in 0 ~ 10 cm soil layer of the
warming sample plot while that in 10 ~ 20 cm soil layer increased slightly and the underground biomass



increased significantly in 20 ~ 30 cm soil layer, which is also related to soil moisture decrease caused by
warming. Since warming reduces soil moisture, water becomes the most key factor limiting plant growth.
In order to better adapt to the environment, plant roots extend to a deeper level for water, thus the
underground biomass transfers to deep soil.
Nitrogen is one of the most needed nutrients (Parras-Alcántara et al., 2013) for plant growth in terrestrial
ecosystem. The addition of nitrogen increases soil mineral nutrients and improves carbon nitrogen ratio
in soil. Moreover, it can improve microbial activity and increase supply of soil respiration, so as to
promote the soil respiration. The result showed that soil respiration rate of nitrogen fertilizer sample plot
was higher than that of the control sample plot. However, statistics showed that the difference was not
significant, indicating no effect of nitrogen in soil respiration.
Though the addition of nitrogen changed the height, density, coverage and frequency as well as
important value of some species of plants, it increased the above-ground biomass of plant species and
plant functional groups, suggesting that the increase of above-ground biomass is not related with the
addition of nitrogen, instead, it may be related to water conditions, soil environment and affinity
interaction of the environment of the year.
Seed rain density of the four main species which make major contribution to plant community was
analyzed. From the perspective the plant community level, we conclude that the seed rain density will not
be significantly affected due to the warming and nitrogen addition process. However, there is significant
difference in years, which is mainly reflected in 2012, 2013 and 2015. From the perspective of species
level, these plants were not significantly affected (expect bassia prostrate), but difference between years
of the four species are significant. For different species, their response to years is different from each
other. The largest seed rain of stipa breviflora occurred in 2014 and 2015 while that of cleistogenes
songorica occurred in 2011, 2013 and 2015. The largest seed rain of allium tenuissimum occurred during
2011-2013 while that of bassia prostrate occurred in 2012, 2013 and 2015. Take the year as the statistic
unit and analyze the impact of interaction effect of warming and nitrogen addition on seed rain density
from 2011 to 2015 (figure 2). Under the condition of warming and nitrogen addition, the seed rain density
of cleistogenes songorica and plant community was not affected. But a significant influence can be found
on allium tenuissimum, bassia prostrate and stipa breviflora in some years. Besides, significant effect
can also be found on the seed rain of allium tenuissimum and bassia prostrate. In addition, when
comparing the impact of warming and nitrogen addition on seed rain of stipa breviflora respectively, a
significant difference is found.
Through the comparison on seed rain of plant community and the fastigium species number of
aboveground vegetation and the analysis of the similarity index applying three-factor analysis of
variance, the results show that aboveground vegetation species number decreased significantly under
the warming condition. However, from the perspective of seed rain species number and shared species
number, there is no significant impact. Still there are significant differences between years (Stott et al.,
2014). The maximum and minimum species number of aboveground vegetation was found in 2012 and
2014 and there is significant difference between them. Besides, there is no significant difference in the
similarity index of Sorense and Jaccard under both warming and nitrogen addition conditions.
**CONCLUSION**



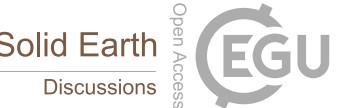

This study takes the stipa breviflora desert grassland in Siziwang Banner, Inner Mongolia as the object
of study, carried out a five-year research by applying the method of man-made warming and artificial
simulation of atmospheric nitrogen settlement, combined with field survey and ecosystem control test.
And an in-depth understanding was obtained through the aspect of soil temperature and humidity,
reproductive phenology of vegetation, as well as seed rain. However, because desert steppe ecosystem
response to climate change involves many aspects and is a process of multi-channel interaction, there
exist some flaws in the results of the research, and there are some factors of uncertainty and ambiguity
as well in the study, which still needs further improvement in future researches.

**ACKNOWLEDGEMENTS**
This study was supported by a grant from the Science and Technology Project of Beijing Municipal
Education Commission (to Sun Qiong) (No. KM201511417009).

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



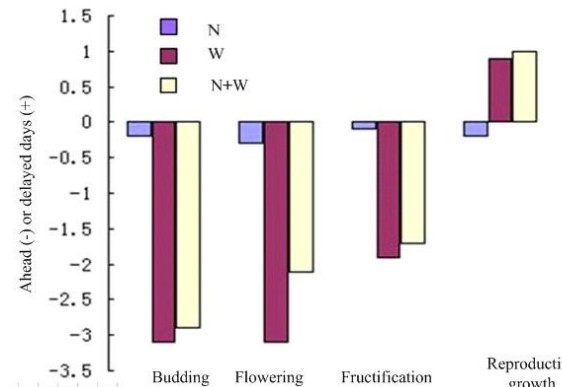

Fig. 1 Impact of warming (W), nitrogen addition (N) and the interaction effect of the two (W+N) on
average time of duration of budding, flowering, fructification and reproductive growth of four dominant
species in the three growing seasons during 2011-2013

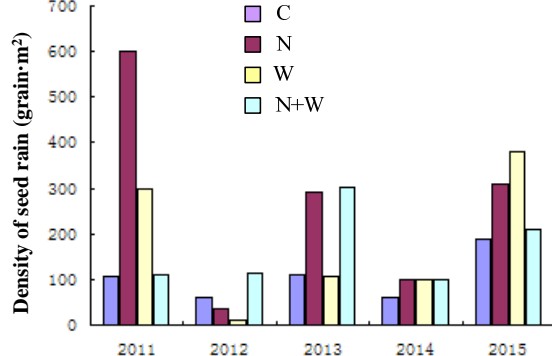


Fig. 2 Impact of warming and nitrogen addition on seed rain density of cleistogenes songorica




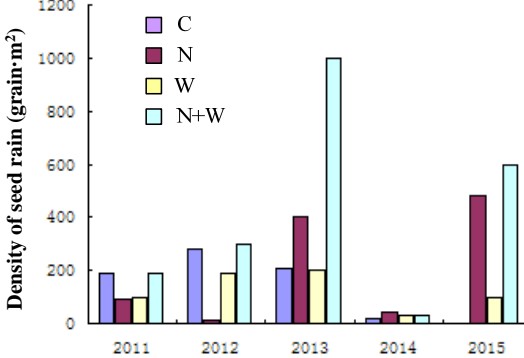

Fig. 3 Impact of warming and nitrogen addition on seed rain density of allium tenuissimum

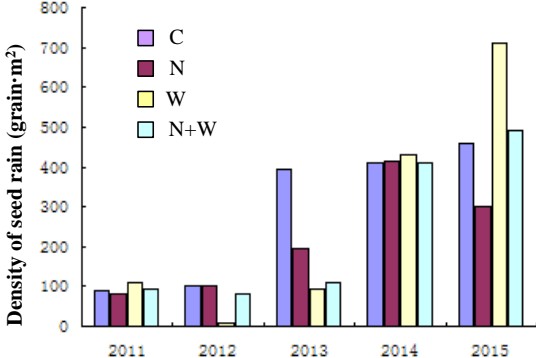

Fig. 4 Impact of warming and nitrogen addition on seed rain density of bassia prostrate


Fig. 5 Impact of warming and nitrogen addition on seed rain density of stipa breviflora





Fig. 6 Impact of warming and nitrogen addition on seed rain density of plant community

Table 1 Two-factor variance analysis results of budding time of the four main species under nitrogen
addtion, warming treatment and the intereaction effect of the two in growing seasons from 2011 to 2013

| Year | Source of variation | plant species | | | |
|------|---------------------|------|------|------|------|
|      |                     | Sb | Ca | At | Kp |
| 2011 | Warming | 0.005 | 0.031 | 0.148 | <0.001 |
|      | Nitrogen | 0.159 | 0.588 | 0.262 | 0.459 |
|      | W*N | 0.309 | 0.194 | 0.062 | 0.153 |
| 2012 | W | 0.008 | 0.231 | 0.004 | 0.166 |
|      | N | 0.666 | 0.546 | 0.474 | 0.959 |
|      | W*N | 0.619 | 0.816 | 0.125 | 0.815 |
| 2013 | W | 0.210 | 0.132 | 0.035 | <0.001 |
|      | N | 0.704 | 0.731 | 0.120 | 0.431 |
|      | W*N | 0.634 | 0.977 | 0.680 | 0.537 |






Table 2 Two-factor variance analysis results of flowering time of four main species under interaction
effect of nitrogen addition and warming in three growing seasons during 2011-2013

| year | Source of variation | plant species | | | |
|---|---|---|---|---|---|
| | | Sb | Ca | At | Kp |
| 2010 | Warming | 0.007 | 0.054 | 0.121 | <0.001 |
| | Nitrogen | 0.148 | 0.783 | 0.411 | 0.126 |
| | W*N | 0.950 | 0.204 | 0.091 | 0.118 |
| 2011 | W | 0.213 | 0.289 | 0.003 | 0.116 |
| | N | 0.471 | 0.229 | 0.512 | 0.485 |
| | W*N | 0.192 | 0.575 | 0.191 | 0.761 |
| 2012 | W | 0.089 | 0.123 | 0.081 | 0.059 |
| | N | 0.845 | 0.286 | 0.671 | 0.163 |
| | W*N | 0.630 | 0.436 | 0.829 | 0.487 |


Table 3 Two-factor variance analysis results of fructification time of four main species under the
interaction effect of warming and nitrogen addition in three growing seasons during 2011 to 2013

| Year | Source of variation | Plant species | | | |
|---|---|---|---|---|---|
| | | Sb | Ca | At | Kp |
| 2011 | Warming | 0.015 | 0.023 | 0.101 | <0.001 |
| | Nitrogen | 0.122 | 0.653 | 0.734 | 0.091 |
| | W*N | 0.343 | 0.262 | 0.172 | 0.112 |
| 2012 | W | 0.983 | 0.556 | 0.012 | 0.240 |
| | N | 0.389 | 0.156 | 0.550 | 0.117 |
| | W*N | 0.083 | 0.923 | 0.520 | 0.013 |
| 2013 | W | 0.015 | 0.811 | 0.089 | 0.632 |
| | N | 0.623 | 0.980 | 0.158 | 0.801 |
| | W*N | 0.711 | 0.836 | 0.739 | 0.881 |




Table 4 Two-factor variance analysis results of reproductive growth duration of four main species under
the interaction effect of warming and nitrogen addition in three growing seasons during 2011-2013

| year | Source of variation | Plant species | | | |
|------|---------------------|------|------|------|------|
| | | Sb | Ca | At | Kp |
| 2011 | Warming | 0.003 | 0.405 | 0.139 | 0.736 |
| | Nitrogen | 0.374 | 0.385 | 0.189 | 0.965 |
| | W*N | 0.054 | 0.959 | 0.871 | 0.545 |
| 2012 | W | 0.009 | 0.269 | 0.359 | 0.291 |
| | N | 0.989 | 0.112 | 0.605 | 0.831 |
| | W*N | 0.379 | 0.602 | 0.125 | 0.511 |
| 2013 | W | 0.105 | 0.037 | 0.653 | <0.001 |
| | N | 0.022 | 0.146 | 0.027 | 0.931 |
| | W*N | 0.604 | 0.545 | 0.222 | 0.139 |



Table 5 The basic information of plant species and seed rain percentage

| Species name | Latin name | Life form | Seed percentage(%) |
|--------------|------------|-----------|---------------------|
| Cleistogenes songorica | Cleistogenes songorica | Perennial hydrophobic cluster grass | 13.11±1 |
| allium tenuissimum | Allium tenuissimum | perennial herb | 37.92±2 |
| Bassia prostrata | Kochia prostrata | subshrub | 14.86±3 |
| Stipa breviflora | Stipabreiflora | Perennial bunch grass | 22.37±2 |
| Neopallasia pectinata | Neopallasiapectinata | Annual and biennial | 2.11±1 |
| Agropyron cristatum | Agropyron cristatum | Perennial clustered | 0.98±0.5 |
| Common Russianthistle | Salsola collina | Annual herb | 7.96±0.3 |
| Convolvulus ammannii | Convoivuius ammannii | perennial herb | 1.12±0.1 |

