# Peer review of "Nitrogen and Warming Control the Vegetation in Inner Mongolia Tourist Area"

_Solid Earth, 2016_

## Referee Comment (RC1) · Dr Oliva (Referee) · 24 Mar 2016

This paper presents an experiment to better understand global warming and atmospheric nitrogen deposition. The study focuses on the Inner Mongolia Autonomous Region and constitutes an interesting approach to examine these topics. However, I have major concerns about the current version of the manuscript.

The structure is not clear. I am afraid that the abstract does not synthesize clearly the methods and findings presented in this study. The reader does not know about the characteristics of the study site, and this is of crucial importance for the objectives of the paper. A need section should be included explaining in detail the main geographical features of the area: climate, geomorphology, geology, vegetation cover, human activities, etc. Also, you should include a (regional/local) map of the area, the reader

needs to know where the study focuses!

Moreover, the results should be better organized. It is very difficult to follow, very dense and poorly structured. Maybe you could organize the results in different subsections. The same for the discussion section, you should split it into different subsections to clearly frame your findings and interpretation. This will make easier for the reader to follow your arguments and interpretation of the entire manuscript. Also, the discussion section could be improved by adding further comparisons with already existing studies on similar topics. I have only counted 5 citations while in the Introduction you refer to several more. What are the most novel findings in your paper with respect to previous studies? Are your results and interpretations similar/different to former studies? What is different in your approach that may be useful for scientists in future studies?

The authors should revise all the citations following the guidelines of the journal. In many cases (e.g. line 43, 45, 46, 52) they mention the name and surnames of the authors and then include the reference at the end. This is not correct. Or they mention the authors at the beginning (Wen & Lu (2015) studied....or only at the end (Wen & Lu, 2015). Check it throughout the text.

Figures and tables are enough and of good quality. But please introduce a new figure 1 (location).

l. 31 please refer to the last IPCC (2013) l. 32 use "°C" here and along the text l. 32 Earth l. 35 increased 1 time with respect to what? l. 35-36 estimated or based? l. 43-55 please rewrite this paragraph citing correctly the references mentioned in the text.

---

## Referee Comment (RC2) · X. Huang (Referee) · 11 Apr 2016

Overall, there are logical problems in this paper. In particular, there is no detailed description on reaction of the vegetation to temperature, humidity and climate as well as the absorption of minerals. If the characteristics of the vegetation itself as well as the difference between the several kinds of vegetation are not well displayed, readers will not be able to understand the changes of the research object properly. Therefore, it is recommended that detailed information on vegetation be supplemented.

Besides, more sufficient argument needs to be added to the discussion part based on the results of the study; some content are mentioned in comparison with other literatures, however, the detailed source of the related literatures or studies is not provided.

Following are the detailed comments the reviewers made:

[Figure]

1. "five year period from March, 2011 to March, 2015" – the data seems confused, please check and make correction. 2. "147757.39 seeds were collected by 64 collectors" - why is the number with decimal place? 3. "The average density of the seed rain is 1229.6±145.8 grain "- what does refer to? 4. As for figure 1, what are the corresponding dominant species to the four set of data, please illustrate it in the figure. 5. For "2010-2012" in figure 4, it was not mentioned in the previous section.

Please also note the supplement to this comment:
http://www.solid-earth-discuss.net/se-2016-52/se-2016-52-RC2-supplement.pdf

---

## Referee Comment (RC3) · H. Liu (Referee) · 13 Apr 2016

This paper studied the effect of warming and nitrogen addition on soil and vegetation effect, which is a new research topic. The outline is clear, however, there are some deficiencies in structure and details.

Some sentences in Results are lengthy and jumbled. It is suggested that the paragraphs of budding and flowering be merged. Discussion part can be simplified by cutting down the sentences or paragraphs irrelevant to the theme of warming and nitrogen addition. It was mentioned that the experiment was carried out on Inner Mongolia Ordos Grassland. The author can add a vegetation map of the experimental area.

Since the experiment base was located in agriculture and pasturage interlaced zone, did the author consider the possible error resulting from the nitrogen in animal

manure? In the Results, the data were mostly based on the experimental results from 2011 to 2013, would the absent data of 2014 and 2015 affect the accuracy of results since it was a five-year experiment from 2011 to 2015? In Discussion, the author mentioned that nitrogen addition had no impact on the soil. The statement is not accurate; more or less, the soil would be affected by nitrogen addition, it is a matter of influence degree. The author also discussed seed rain with which the readers might not be familiar. It would be better if the author explain the concept of seed rain.

Please also note the supplement to this comment:
http://www.solid-earth-discuss.net/se-2016-52/se-2016-52-RC3-supplement.pdf